Acute psycho-physiological responses to submaximal constant-load cycling under intermittent hypoxia-hyperoxia vs. hypoxia-normoxia in young males

Behrendt Tom 1 tom.behrendt@ovgu.de
Bielitzki Robert 1
Behrens Martin 2
Jahns Lina-Marie 1
Boersma Malte 1
Schega Lutz 1
1 Department of Sport Science, Chair for Health and Physical Activity, Otto-von-Guericke University Magdeburg , Magdeburg , Germany
2 University of Applied Sciences for Sport and Management Potsdam , Potsdam , Germany
Jimenez Manuel
Electronic publication date: 2024 Oct 4
Publication date: 2024
Volume: 12
Electronic Location ID: e18027
Received 2024 Mar 22; Accepted 2024 Aug 12
Copyright: © 2024 Behrendt et al.
Copyright year: 2024
Copyright holder: Behrendt et al.
License: This is an open access article distributed under the terms of the Creative Commons Attribution License, which permits unrestricted use, distribution, reproduction and adaptation in any medium and for any purpose provided that it is properly attributed. For attribution, the original author(s), title, publication source (PeerJ) and either DOI or URL of the article must be cited.
License URL: https://creativecommons.org/licenses/by/4.0/

Keywords: Perceived motor fatigue, Effort perception, Perceived physical strain, Exercise enjoyment, Near-infrared spectroscopy, Blood lactate

Funding: The authors received no funding for this work.

==============================
Background

Hypoxia and hyperoxia can affect the acute psycho-physiological response to exercise. Recording various perceptual responses to exercise is of particular importance for investigating behavioral changes to physical activity, given that the perception of exercise-induced pain, discomfort or unpleasure, and a low level of exercise enjoyment are commonly associated with a low adherence to physical activity. Therefore, this study aimed to compare the acute perceptual and physiological responses to aerobic exercise under intermittent hypoxia-hyperoxia (IHHT), hypoxia-normoxia (IHT), and sustained normoxia (NOR) in young, recreational active, healthy males.

Methods

Using a randomized, single-blinded, crossover design, 15 males (age: 24.5 ± 4.2 yrs) performed 40 min of submaximal constant-load cycling (at 60% peak oxygen uptake, 80 rpm) under IHHT (5 × 4 min hypoxia and hyperoxia), IHT (5 × 4 min hypoxia and normoxia), and NOR. Inspiratory fraction of oxygen during hypoxia and hyperoxia was set to 14% and 30%, respectively. Heart rate (HR), total hemoglobin (tHb) and muscle oxygen saturation (SmO2) of the right vastus lateralis muscle were continuously recorded during cycling. Participants’ peripheral oxygen saturation (SpO2) and perceptual responses (i.e., perceived motor fatigue, effort perception, perceived physical strain, affective valence, arousal, motivation to exercise, and conflict to continue exercise) were surveyed prior, during (every 4 min), and after cycling. Prior to and after exercise, peripheral blood lactate concentration (BLC) was determined. Exercise enjoyment was ascertained after cycling. For statistical analysis, repeated measures analyses of variance were conducted.

Results

No differences in the acute perceptual responses were found between conditions (p ≥ 0.059, ηp2 ≤ 0.18), while the physiological responses differed. Accordingly, SpO2 was higher during the hyperoxic periods during the IHHT compared to the normoxic periods during the IHT (p < 0.001, ηp2 = 0.91). Moreover, HR (p = 0.005, ηp2 = 0.33) and BLC (p = 0.033, ηp2 = 0.28) were higher during IHT compared to NOR. No differences between conditions were found for changes in tHb (p = 0.684, ηp2 = 0.03) and SmO2 (p = 0.093, ηp2 = 0.16).

Conclusion

IHT was associated with a higher physiological response and metabolic stress, while IHHT did not lead to an increase in HR and BLC compared to NOR. In addition, compared to IHT, IHHT seems to improve reoxygenation indicated by a higher SpO2 during the hyperoxic periods. However, there were no differences in perceptual responses and ratings of exercise enjoyment between conditions. These results suggest that replacing normoxic by hyperoxic reoxygenation-periods during submaximal constant-load cycling under intermittent hypoxia reduced the exercise-related physiological stress but had no effect on perceptual responses and perceived exercise enjoyment in young recreational active healthy males.

Introduction

Portions of this text were previously published as part of a preprint (https://www.researchsquare.com/article/rs-3786050/v1).

Regular physical activity has been recognized as a valuable strategy to promote health and to prevent or treat chronic disease (Pareja-Galeano, Garatachea & Lucia, 2015; Pedersen & Saltin, 2015; Ruegsegger & Booth, 2018). The World Health Organization recommended that individuals aged between 18 and 64 years should undertake moderate-intensity aerobic activity for a minimum of 150 min per week for substantial health benefits (e.g., reduced all-cause mortality, symptoms of anxiety and depression, as well as incident of hypertension and type 2 diabetes mellitus) (Bull et al., 2020).

To date, several studies have investigated the effect of aerobic training under hypoxia on health- and performance-related outcomes in nonathletic, sedentary, and clinical populations (Lizamore & Hamlin, 2017; Camacho-Cardenosa et al., 2019; Millet et al., 2016). Although moderate-intensity aerobic training under hypoxic conditions does not appear to be more effective than a similar training in normoxia in improving exercise capacity, the effects on vascular health and cardiometabolic risk factors are promising (Lizamore & Hamlin, 2017; Montero & Lundby, 2016; Ramos-Campo et al., 2019). In this context, Muangritdech et al. (2020) showed that 6 weeks of breathing intermittent hypoxia (eight cycles at 3 min hypoxia (fraction of inspired oxygen (FiO2) = 0.141) and 3 min normoxia) in combination with submaximal aerobic exercise (i.e., 48 min of continuous treadmill-walking at 35–50% of individuals’ heart rate reserve) was effective in reducing systolic and diastolic blood pressure as well as in increasing the 6 min-walk-test performance in hypertensive patients. Moreover, breathing intermittent hypoxia in combination with exercise was more effective than breathing intermittent hypoxia at rest indicated by higher blood-levels of nitric oxide metabolites as well as hypoxia-inducible factor-1α and a lower malondialdehyde level two days after the intervention period (Muangritdech et al., 2020).

In the last decades, a new hypoxic method was developed combining hypoxic and hyperoxic periods to increase the adaptive response to the intermittent hypoxic stimulus by upregulating reactive oxygen species (ROS) and hypoxia-inducible genes (Mallet et al., 2020), as shown in animal studies (Gonchar & Mankovska, 2012; Arkhipenko, Sazontova & Zhukova, 2005; Sazontova et al., 2012). The results of recent studies suggest that interventions including intermittent hypoxic-hyperoxic exposure at rest (IHHE) are efficient to improve or even maintain cognitive and physical performance, as well as metabolic and cardiovascular parameters in clinical populations (Behrendt et al., 2022). However, so far, only one study has directly compared the chronic effects of exposure to intermittent hypoxia-normoxia and IHHE on humans’ health and/or performance (Serebrovska et al., 2019). In this regard, Serebrovska et al. (2019) found no advantages of IHHE over intermittent hypoxic-normoxic exposure on blood glucose levels in prediabetic patients. Accordingly, there is no clear evidence to support the hypothesis that replacing normoxic by hyperoxic periods can increase the effectiveness of hypoxia interventions in humans. In addition, to the best of the authors’ knowledge, there is only one study so far that has investigated the acute effect of intermittent hypoxia-hyperoxia in combination with exercise (i.e., intermittent hypoxic-hyperoxic training (IHHT)) (Brinkmann et al., 2017). Therefore, the health- and performance-related effects of IHHT are largely unknown. In a pilot study, Brinkmann et al. (2017) reported that breathing intermittent hypoxic-hyperoxic (eight cycles at 5 min hypoxia (FiO2 = 0.140) and 5 min hyperoxia (FiO2 = 0.300)) and sustained hypoxic air (FiO2 = 0.140) in combination with submaximal constant-load cycling (40 min at an load equivalent to that was determined at a blood lactate concentration of 2.5 mmol/l) acutely increased vascular endothelial growth factor and matrix metalloproteinases-2 blood level in overweight/obese patients with type 2 diabetes mellitus, whereas cycling in normoxia did not increase these pro-angiogenic factors. Moreover, although not statistically significant, authors found that most patients (seven of nine) rated perceived exertion (15-point (6–20) Borg scale) to be lower during IHHT compared with cycling under sustained hypoxia. Collectively, this implies that combining hypoxic and hyperoxic periods during cycling is particularly beneficial for overweight/obese patients with type 2 diabetes mellitus as it can increase angiogenic regulators similar to sustained hypoxic exposure, but seems to be better tolerated by most patients (Brinkmann et al., 2017). In this context, recording various perceptual responses to exercise related to different dimensions (i.e., sensory-discriminatory, affective-motivational, and cognitive-evaluative dimension) (Venhorst, Micklewright & Noakes, 2018) is of particular importance for investigating physical activity behaviour changes towards health, given that the perception of exercise-induced pain, discomfort or unpleasure, and a low level of exercise enjoyment are commonly associated with a low adherence to physical activity (Ekkekakis & Zenko, 2016; Brand & Ekkekakis, 2018; Rhodes & Kates, 2015). There is a lack of studies thoroughly examining the perceptual responses belonging to the sensory-discriminatory, affective-motivational, cognitive-evaluative dimension, and exercise enjoyment during and after IHHT, respectively.

Therefore, the aim of this study was to compare the acute psycho-physiological responses to 40 min of cycling at a constant load corresponding to 60% of individuals’ peak oxygen uptake (V̇O2peak) under intermittent hypoxia-normoxia, hypoxia-hyperoxia, and sustained normoxia. For this purpose, acute perceptual (perceived motor fatigue, effort perception, perceived physical strain, affective valence, arousal, motivation to exercise, conflict to continue exercise, and enjoyment) and physiological responses (heart rate, blood lactate concentration, muscle oxygenation, and peripheral oxygen saturation) were measured before, during, and/or after a single exercise session in each condition. It was hypothesized that submaximal constant-load cycling under hypoxia-hyperoxia induces lower ratings in perceived motor fatigue, effort perception, perceived physical strain, arousal, and conflict to continue exercise as well as higher ratings in affective valence, motivation to exercise, and enjoyment, accompanied by lower heart rate, blood lactate concentration, as well as greater muscle oxygenation and peripheral oxygen saturation (SpO2) compared to hypoxia-normoxia.

Materials and Methods

Experimental design

All participants attended the laboratory on four separate days (cardiopulmonary exercise test and three experimental trials, see Fig. 1). At the first visit, participants’ eligibility was determined and anthropometric data were documented. Participants were also familiarized with the experimental procedure including the explanation of the rating scales (i.e., perceived motor fatigue, effort perception, perceived physical strain, affective valence, arousal, motivation to exercise, and conflict to continue exercise). Furthermore, participants performed a cardiopulmonary exercise test on a bicycle ergometer. In a randomized, counterbalanced cross-over design, participants completed three experimental trials at a similar time of the day to reduce circadian influence, separated by at least 4 days. Upon arrival, participants completed a free German version of the Profile of Mood States questionnaire (POMS) (McNair, Lorr & Droppleman, 1992; Dalbert, 1992; Leibniz-Institut für Psychologie (ZPID), 2019) to check for differences in trait mood (queried over the course of the last 7 days) between conditions. A standardized 5-min warm-up was performed on a cycle ergometer at an individual load corresponding to 40% of participants’ V̇O2peak and a cadence of 80 rpm. Thereafter, participants performed 40 min of submaximal cycling at a constant-load corresponding to 60% of participants’ V̇O2peak and a cadence of 80 rpm either under (i) intermittent hypoxic-normoxic (IHT: 4 min of hypoxia (FiO2 ≈ 0.140) interspersed by 4 min of normoxia (FiO2 ≈ 0.209) for five cycles), (ii) intermittent hypoxic-hyperoxic (IHHT: 4 min of hypoxia (FiO2 ≈ 0.140) interspersed by 4 min of hyperoxia (FiO2 ≈ 0.300) for five cycles), or (iii) sustained normoxic conditions (NOR: FiO2 ≈ 0.209).

Figure 1 Study design and experimental procedure.

BSL, baseline; FiO2, inspiratory fraction of oxygen; IHHT, intermittent hypoxic-hyperoxic training condition; IHT, intermittent hypoxic-normoxic training condition; NOR, normoxic training condition; PACES, physical activity enjoyment scale; POMS, profile of mood states; V̇O2peak, peak oxygen consumption. Icons from PowerPoint. Figure created with PowerPoint.

Heart rate and muscle oxygenation of the right vastus lateralis were continuously recorded 1 min prior and during cycling. Perceived motor fatigue, effort perception, perceived physical strain, affective valence, arousal, motivation to exercise, and conflict to continue exercise were queried after 4 (T4), 8 (T8), 20 (T20), 24 (T24), 36 (T36), and 40 min (T40) of submaximal constant-load cycling in each condition. Furthermore, perceived motor fatigue, affective valence, and arousal were also queried prior to (T0) and 10 min after (post 10) the 40-min cycling trial, whereas motivation to exercise was additionally assessed only at T0. Prior to (BSL) as well as immediately (post 0) and 10 min (post 10) after the 40-min cycling trial, peripheral blood lactate concentration (BLC) was determined. Participants’ SpO2 was measured prior to (BSL) and after every 4 min period during the 40-min cycling trial (i.e., at 4 (P4), 8 (P8), 12 (P12), 16 (P16), 20 (P20), 24 (P24), 28 (P28), 32 (P32), 36 (P36), and 40 min (P40)). Furthermore, participants were asked to complete a revised 16-item version of the physical activity enjoyment scale (PACES) shortly after the end of cycling to measure exercise enjoyment. Ambient temperature was measured for each experimental trial (Thermohygrometer 608-H1; Testo SE & Co. KGaA, Germany).

Participants

A sample size calculation was performed for a 3 (condition: IHT, IHHT, normoxia) × 8 (time: baseline, after 4, 8, 20, 24, 36, and 40 min as well as post 10 min) repeated measures analysis of variance (ANOVA) with physical strain as the primary outcome. However, due to the lack of studies comparing the acute effect of breathing intermittent hypoxia-normoxia and hypoxia-hyperoxia during submaximal constant-load cycling on psycho-physiological responses a sample size calculation was not possible on this basis. Evidence from previous research suggest that there is a medium to large effect (Cohens’ f = 0.25 to 0.80) of sustained hypoxia or hyperoxia vs. normoxia on acute perceptual responses (i.e., physical strain) during maximal and submaximal intermittent and constant-load cycling in trained runners (Hobbins et al., 2019a) and cyclists (Sperlich et al., 2012) as well as patients with chronic obstructive pulmonary disease (O’Donnell, D’Arsigny & Webb, 2001). Thus, a medium effect size (Cohens’ f = 0.25) with a significance level of α = 0.05, a power of 1-β = 0.80, and an expected correlation between measures of 0.7 were used as input variables for the sample size calculation.

Accordingly, a total sample size of 15 physically active healthy males aged between 20 and 39 years were recruited (age: 24.5 ± 4.2 years, height: 1.81 ± 0.07 m, weight: 80.3 ± 9.6 kg, body mass index: 24.4 ± 2.3 kg/m2, peak oxygen consumption [V̇O2peak]: 4.1 ± 0.6 L/min1, 51.4 ± 6.5 ml/min1/kg1). Participants were eligible for recruitment if they met the following criteria: (i) no musculoskeletal injuries within the past 6 months, (ii) no acute pain or illness, (iii) free from clinical signs of orthopedic, neurological, cardiovascular, or respiratory disease or injuries, (iv) no drug intake with nervous, cardiovascular, or respiratory impact, and (v) no exposure to hypoxia (≥ 2,000 m) for ≥ 48 h within the last 6 months. Furthermore, participants were instructed to refrain from vigorous exercise for 48 h, to avoid alcohol and caffeine for 24 h, and to take their last meal ≥ 2 h before each visit. Written informed consent was obtained from all participants. The study was approved by the local Ethics Committee of the Otto-von-Guericke University Magdeburg, Germany (approval number: 68/23) and confirmed the principles of the Declaration of Helsinki on human experimentation.

Hypoxic and hyperoxic application

Normobaric hypoxic, hyperoxic, and normoxic air were applied via a facemask connected through flexible plastic tubes to a hypoxic generator (Everest Summit II, Hypoxico, New York, NY, USA) and an oxygen container (10 l medical oxygen (FiO2 = 0.999); GTI medicare, Hattingen, Germany). During the IHHT and IHT conditions, participants started cycling under hypoxia (FiO2 ≈ 0.140) for 4 min. Thereafter, the hypoxic generator was switched by the investigator from hypoxia to normoxia (FiO2 ≈ 0.209) and vice versa every 4 min until a total of five cycles were achieved. Solely in the IHHT condition, oxygen was manually admixed to the normoxic air to induce hyperoxia (FiO2 ≈ 0.300). The FiO2 of the supplied air mixture was continuously monitored by means of an oxygen sensor (GOX 100; GHM Group—Greisinger, Regenstauf, Germany) and was individually adjusted if necessary. During the NOR condition, participants were breathing normoxic air through a facemask connected to the hypoxic generator. Participants were blinded to each condition as the hypoxia generator was working during each experimental trial with the FiO2 not visible.

Measurements

Cardiopulmonary exercise test

To determine the individual V̇O2peak and power output at 40% and 60% V̇O2peak, participants performed a cardiopulmonary exercise test. For this, an incremental load test (initial load: 50 W, increment: 50 W every 3 min until volitional exhaustion, cadence: 80 rpm) was performed under normoxic condition on a bicycle ergometer (Xrcise Cycle Med; Cardiowise, Großsteinhausen, Germany) in combination with spirometry (MetaMax 3B R2; CORTEX Biophysik, Leipzig, Germany) and heart rate monitoring (H7; Polar®, Finland). The exercise test was terminated upon volitional exhaustion (i.e., inability to maintain pedal frequency) or if the participant requested termination of the test. All participants reached a maximal respiratory exchange ratio of ≥ 1.17 and ≥ 93% of their age predicted maximum heart rate (210—age, formula was adapted considering the lower muscle mass involved during cycling (Knaier et al., 2019)). Gas exchange measurement was conducted breath-by-breath. Data were averaged over 45 s and V̇O2peak was defined as the highest mean V̇O2 during 30 s of measurement. Absolute (l/min1) and relative V̇O2peak (ml/kg1/min1) were determined.

Rating scales

Prior to each experimental trial, participants were given standardized instructions on the meaning of each rating scale. Perceived motor fatigue was queried by asking “How fatigued do you feel currently?” via Borg’s category ratio 10 scale ranging from 0 (“nothing at all”) to 10 (“maximally fatigued”) (Micklewright et al., 2017). Effort perception was queried by asking “How hard is it for you to drive your legs?” based on a 15-point Borg scale ranging from 6 (“not hard at all”) up to 20 (“maximum hard”). Participants were instructed that the “effort perception” scale is used to evaluate the level of subjective awareness of effort invested into or required to continue with the task (Venhorst, Micklewright & Noakes, 2018). With regard to perceived physical strain, participants were instructed to evaluate the intensity of their exercise-related sensations such as muscular, respiratory, and thermal strain caused by the task based on a 15-point Borg scale ranging from 6 (“barely perceptible”) up to 20 (“maximum strong”) (Venhorst, Micklewright & Noakes, 2018). Affective valence was assessed using the single-item dimensional feeling scale and queried by asking “How are you feeling right now?” via a 11-point scale ranging from −5 (“very bad”) up to +5 (“very good”) (Ekkekakis & Russell, 2013). Arousal was queried by asking “How aroused do you feel right now?” via 6-point felt arousal scale ranging from 1 (“low arousal”) up to 6 (“high arousal”) (Ekkekakis & Russell, 2013). Motivation to exercise was queried by asking “How motivated do you feel right now?” via a 20 cm visual analog scale ranging from 0 (“not at all”) up to 10 (“extremely high”). To assess the perceived conflict to continue exercise (i.e., action crisis), participants were asked to rate to which extent they were seriously conflicted whether to continue or to stop exercising via a 20 cm visual analog scale ranging from 0 (“not applicable at all”) up to 10 (“fully applies”) (Brandstätter & Schüler, 2013). During submaximal constant-load cycling, each scale was presented by the investigator in front of the participant on a board (21.0 × 29.7 cm). Scale order presentation was held constant across experimental trials and participants.

The modified PACES was used (i.e., PACES-16) to assess enjoyment experienced during the experimental trials (Motl et al., 2001). The modified PACES consists of 16-items, which have to be rated on a 5-point Likert-type scale ranging from 1 (“Disagree a lot”) to 5 (“Agree a lot”) (Motl et al., 2001; Jekauc et al., 2013). The total score of the PACES-16 ranges between 16 and 80, with higher scores representing greater enjoyment.

Muscle oxygenation

Muscle oxygenation was assessed using a muscle near-infrared spectroscopy (mNIRS) monitor (MOXY; Fortiori Design LLC, Hutchinson, MN, USA), which recorded changes in total tissue hemoglobin concentration (tHb) and oxygenated hemoglobin as a percentage of tHb (i.e., muscle oxygenation saturation, SmO2) (Crum et al., 2017). The MOXY device was placed on the muscle belly of the right vastus lateralis half distance between the patella base and the greater trochanter. The application area was shaved and cleaned with disinfectant. Thereafter, a light protecting cap (ø = 125 mm) was attached around the mNIRS monitor and fixed with adhesive tape. SmO2 and tHb were recorded throughout the 40-min cycling trial with a sampling rate of 0.5 Hz and filtered with a 4th order low-pass zero-phase Butterworth filter with a cutoff frequency of 0.2 Hz (Husmann et al., 2019). Data were averaged over the last 120 s of every 4-min period (i.e., at 4 (P4), 8 (P8), 12 (P12), 16 (P16), 20 (P20), 24 (P24), 28 (P28), 32 (P32), 36 (P36), and 40 min (P40)) and normalized to baseline (60 s) recorded immediately before exercise in a seated position on the cycle ergometer (i.e., percentage changes to baseline: %ΔSmO2 and %ΔtHb).

Blood lactate concentration, peripheral oxygen saturation, and heart rate

Capillary blood samples (10 µl) were collected from the right hyperemic earlobe. Samples were stored in a glucocapil reaction cup, prefilled with 1,000 µl hemolysate-system-solution to hemolyze and dilute samples for BLC determination using a device that runs the enzymatic amperometric method (Super GL3; Hitado GmbH, Möhnesee, Germany, measuring error < 1.5%). SpO2 was recorded using a fingertip pulse oximeter (Pulox PO-200; Novidion GmbH, Cologne, Germany) attached to the right index finger with a covert display making it not visible to the participants. Heart rate was recorded continuously using a heart rate strap (H10; Polar Electro Oy, Pohjois-Pohjanmaa, Finland) and monitor (Vintage V2; Polar Electro Oy, Finland). Raw data were transferred to a PC using the software “Polar FlowSync” (Polar Electro Oy, Pohjois-Pohjanmaa, Finland). The Kubios HRV software (version 3.5.0; Kubios Oy, Finland) was used to detect artefacts and for further data processing (Tarvainen et al., 2014). R-R-intervals that deviated by more than 30% from the previous interval were classified as artefacts and replaced by the average calculated from the previous and subsequent value (Gronwald, Hoos & Hottenrott, 2019). Data were averaged over 60 s for baseline recording (BSL) and over every 4-min period during cycling (i.e., at 4 (P4), 8 (P8), 12 (P12), 16 (P16), 20 (P20), 24 (P24), 28 (P28), 32 (P32), 36 (P36), and 40 min (P40)).

Statistical analysis

Descriptive data are shown as means ± standard deviations (SD). In- and between-group differences are reported as mean differences (MD) with 95% confidence intervals (95% CI). The SpO2 to FiO2 ratio (SF-index = SpO2 ÷ FiO2) was calculated to present participants’ individual SpO2 response to hypoxia (Soo et al., 2020). Statistical analysis was performed using JASP Statistics (Version 0.16.2, University of Amsterdam, Netherlands). One-way (condition) ANOVAs were performed for trait mood (i.e., POMS), exercise enjoyment (i.e., PACES), and ambient temperature. Two-way (condition × time) repeated measures ANOVAs were conducted for all remaining parameters. In case of sphericity violation, Greenhouse-Geisser correction was applied. Since it was shown that ANOVA and t-test could be used without significant error despite violation of homogeneity of variance and normal distribution (Schmider et al., 2010; Havlicek & Peterson, 1974), the data were not checked for these assumptions and no alternative nonparametric tests were performed. In case of significant interactions or main effects, post-hoc tests with Bonferroni corrections were conducted. The effect sizes partial eta squared (ηp2: small ≥ 0.01, medium ≥ 0.06, large ≥ 0.14 (Lakens, 2013)) and Cohens’ d (d: small ≥ 0.10, medium ≥ 0.50, large ≥ 0.80 (Cohen, 1992)) were calculated for the ANOVAs and post-hoc tests, respectively. The level of significance was set at p < 0.050.

Results

All participants successfully completed the experimental trials without adverse events. There were no differences between conditions in participants’ trait mood level (i.e., POMS) as well as in ambient temperature (Table 1).

Table 1 Participants’ trait mood state, exercise enjoyment, and ambient temperature recorded during the experimental conditions.

Values are presented as means ± standard deviations. ANOVA, analysis of variance; IHHT, intermittent hypoxia-hyperoxia; IHT, hypoxia-normoxia; NOR, sustained normoxia.

Value	IHHT	IHT	NOR	One-way repeated measures ANOVA	
Profile of mood state	61.2 ± 8.0	62.6 ± 8.1	60.7 ± 7.3	F1.381,19.329 = 0.472, p = 0.561, ηp2 = 0.03	
Physical activity enjoyment scale	59.0 ± 11.2	58.4 ± 10.7	60.1 ± 11.3	F2, 26 = 0.467, p = 0.632, ηp2 = 0.04	
Ambient temperature (°C)	24.4 ± 1.3	24.2 ± 1.1	24.1 ± 0.9	F2, 28 = 0.343, p = 0.712, ηp2 = 0.02	

Rating scales

Descriptive values and results of the repeated measures ANOVAs for perceived motor fatigue, effort perception, physical strain, affective valence, arousal, motivation, and conflict to continue exercise are presented in Fig. 2 and Table 2, respectively.

Figure 2 Perceived motor fatigue (A), affective valence (B), arousal (C), motivation to exercise (D), conflict to continuous exercise (E), perceived physical strain (F), and effort perception (G).

Perceptual responses were quired immediately before (BSL) as well as after 4 (T4), 8 (T8), 20 (T20), 24 (T24), 36 (T36), and 40 min (T40) during submaximal constant-load cycling under either intermittent hypoxia-hyperoxia (IHHT), intermittent hypoxia-normoxia (IHT), or sustained normoxia (NOR). Values are presented as means and standard deviations.

Table 2 Perceptual responses prior to (T0), during (at 4 (T4), 8 (T8), 20 (T20), 24 (T24), 36 (T36), and 40 min (T40)), and 10 min after (post 10) exercise.

Values are presented as means ± standard deviations. ANOVA, analysis of variance; IHHT, intermittent hypoxia-hyperoxia; IHT, hypoxia-normoxia; NOR, sustained normoxia.

Condition	T0	T4	T8	T20	T24	T36	T40	Post 10	Repeated measures ANOVA	
Time × condition interaction	Main effect of time	Main effect of condition	
Perceived motor fatigue	
IHHT	0.7 ± 1.1	2.6# ± 2.2	2.6# ± 1.7	4.5# ± 1.7	3.5# ± 1.3	4.9# ± 1.7	4.1# ± 1.4	1.9 ± 1.4	F14,182 = 2.233, p = 0.009, ηp2 = 0.15	F2.509, 32.622 = 40.478, p < 0.001, ηp2 = 0.76	F2, 26 = 0.191, p = 0.823, ηp2 = 0.02	
IHT	0.9 ± 1.1	2.8# ± 1.6	2.4 ± 1.4	4.5# ± 1.6	3.6# ± 1.3	5.1# ± 1.7	4.5# ± 1.7	2.1 ± 1.4	
NOR	1.2 ± 1.1	1.8 ± 1.5	2.5 ± 1.7	3.9# ± 1.7	4.2# ± 1.8	4.6# ± 1.7	4.7# ± 1.4	2.0 ± 1.1	
Effort perception	
IHHT		11.1 ± 3.4	10.7 ± 2.5	13.2# ± 2.8	11.9 ± 2.3	14.1# ± 2.3	12.5 ± 1.8		F5.064, 70.902 = 2.728, p = 0.026, ηp2 = 0.16	F2.083, 29.164 = 30.063, p < 0.001, ηp2 = 0.68	F2, 28 = 3.145, p = 0.059, ηp2 = 0.18	
IHT		10.9 ± 2.7	10.1 ± 2.1	12.8# ± 2.7	11.8 ± 2.0	14.1# ± 2.8	12.8# ± 2.6		
NOR		9.7 ± 2.5	10.0 ± 2.7	11.7# ± 2.2	11.9# ± 2.4	12.5# ± 2.0	12.7# ± 2.2		
Physical strain	
IHHT		9.9 ± 2.2	9.8 ± 2.0	12.4# ± 2.4	11.2 ± 2.6	13.1# ± 2.4	11.9# ± 1.8		F10, 140 = 3.348, p < 0.001, ηp2 = 0.19	F2.179, 30.509 = 41.621, p < 0.001, ηp2 = 0.75	F2, 28 = 1.950, p = 0.164, ηp2 = 0.12	
IHT		10.1 ± 2.3	9.9 ± 2.0	12.4# ± 2.4	11.5 ± 1.7	13.7# ± 2.6	12.3# ± 2.4		
NOR		8.7 ± 2.0	9.6 ± 2.3	11.4# ± 2.2	11.7# ± 2.2	12.3# ± 2.1	12.7# ± 2.2		
Affective valence	
IHHT	3.2 ± 1.6	2.4 ± 2.3	3.1 ± 1.3	2.1 ± 1.5	2.8 ± 1.5	2.1 ± 1.6	3.1 ± 1.5	4.1 ± 1.1	F14, 182 = 1.885, p = 0.030, ηp2= 0.13	F3.261, 42.388 = 11.905, p < 0.001, ηp2 = 0.48	F2, 26 = 1.181, p = 0.320, ηp2 = 0.08	
IHT	3.3 ± 2.0	2.6 ± 2.1	3.3 ± 1.5	2.6 ± 1.9	2.9 ± 1.7	2.1 ± 1.8	3.0 ± 1.7	4.1 ± 0.8	
NOR	3.6 ± 1.4	3.3 ± 1.5	2.9 ± 1.6	2.8 ± 1.5	2.8 ± 1.6	3.1 ± 1.2	3.6 ± 1.2	4.0 ± 1.0	
Arousal	
IHHT	1.8 ± 1.2	2.5 ± 1.1	2.3 ± 1.0	3.0# ± 0.8	2.8 ± 0.9	3.2# ± 1.1	3.0# ± 0.7	1.9 ± 1.0	F14, 168 = 0.924, p = 0.534, ηp2 = 0.07	F2.060, 24.724 = 12.531, p < 0.001, ηp2 = 0.51	F2, 24 = 0.178, p = 0.838, ηp2 = 0.02	
IHT	1.6 ± 1.0	2.4 ± 1.1	2.4 ± 1.1	3.2# ± 1.2	2.8# ± 1.0	3.5# ± 1.5	3.3# ± 1.1	2.1 ± 1.3	
NOR	1.8 ± 1.0	2.2 ± 1.1	2.5 ± 1.1	2.8 ± 1.0	3.0# ± 1.1	3.2# ± 1.1	3.1# ± 1.1	1.9 ± 1.0	
Motivation to exercise	
IHHT	7.9 ± 2.1	7.1 ± 2.5	7.9 ± 1.9	7.5 ± 2.1	7.7 ± 2.1	7.3 ± 2.1	8.1 ± 1.4		F4.680, 65.527 = 1.472, p = 0.214, ηp2 = 0.10	F2.374, 33.241 = 1.778, p = 0.179, ηp2 = 0.11	F2, 28 = 0.262, p = 0.771, ηp2 = 0.02	
IHT	8.1 ± 2.5	7.5 ± 2.6	7.9 ± 2.3	7.7 ± 2.3	8.0 ± 2.1	7.3 ± 2.4	8.1 ± 2.1		
NOR	8.1 ± 1.9	8.0 ± 1.9	7.8 ± 2.1	7.5 ± 2.1	7.5 ± 2.1	8.0 ± 1.6	8.0 ± 1.6		
Conflict to continue exercise	
IHHT		1.7 ± 2.5	1.2 ± 1.7	2.0 ± 1.8	1.3 ± 1.7	1.3 ± 2.0	0.3 ± 0.7		F4.640, 64.966 = 2.357, p = 0.054, ηp2 = 0.14	F1.347, 18.854 = 2.561, p = 0.119, ηp2 = 0.16	F2, 28 = 2.182, p = 0.132, ηp2 = 0.14	
IHT		2.0 ± 2.5	1.4 ± 2.0	1.7 ± 2.1	1.2 ± 1.7	1.5 ± 2.1	0.8 ± 1.7		
NOR		0.9 ± 1.8	1.2 ± 1.9	1.5 ± 1.7	1.3 ± 1.4	0.5 ± 1.2	0.4 ± 0.8		
Note:

# denotes a difference (p < 0.05) compared to T0 (i.e., perceived motor fatigue, affective valence, arousal, motivation to exercise) or T1 (i.e., physical strain, effort perception, and conflict of continue exercise).

Perceived motor fatigue: There was a time × condition interaction and a main effect of time but no main effect of condition for perceived motor fatigue. Post-hoc tests revealed no differences between conditions. Perceived motor fatigue was higher at T4 compared to T0 during IHHT (MD = 1.9 (95% CI [0.3–3.4], p = 0.002, d = 1.22) and IHT (MD = 1.9 (95% CI [0.4–3.5]), p < 0.001, d = 1.27). Perceived motor fatigue was further increased compared to T0 at T8 (MD = 1.9 (95% CI [0.4–3.5]), p < 0.001, d = 1.27), T20 (MD = 3.8 (95% CI [2.3–5.3]), p < 0.001, d = 2.49), T24 (MD = 2.8 (95% CI [1.3–4.3]), p < 0.001, d = 1.83), T36 (MD = 4.1 (95% CI [2.6–5.7]), p < 0.001, d = 2.72), and T40 (MD = 1.9 (95% CI [0.4–3.5]), p < 0.001, d = 1.27) during IHHT and at T20 (MD = 3.6 (95% CI [2.1–5.4]), p < 0.001, d = 2.39), T24 (MD = 2.8 (95% CI [1.3–4.3]), p < 0.001, d = 1.83), T36 (MD = 4.3 (95% CI [2.8–5.8]), p < 0.001, d = 2.81), and T40 (MD = 3.6 (95% CI [2.1–5.3]), p < 0.001, d = 1.39) during IHT. During NOR, perceived motor fatigue was higher at T20 (MD = 2.6 (95% CI [1.1–4.2]), p < 0.001, d = 1.74), T24 (MD = 3.0 (95% CI [1.5–4.3]), p < 0.001, d = 1.97), T36 (MD = 3.4 (95% CI [1.8–4.9]), p < 0.001, d = 2.20), and T40 (MD = 3.5 (95% CI [2.0–5.0]), p < 0.001, d = 2.30) compared to T0.

Effort perception: There was a time × condition interaction and main effect of time but no main effect of condition for effort perception. Post-hoc tests revealed no differences between conditions. Effort perception was higher at T20 (IHHT: MD = 2.1 (95% CI [3.7–0.4]), p = 0.002, d = 0.837; IHT: MD = 1.9 (95% CI [3.5–0.2]), p = 0.009, d = 0.76; CON: MD = 2.0 (95% CI [3.7–0.3]), p = 0.003, d = 0.81) and T36 (IHHT: MD = 32 (95% CI [4.7–1.3]), p < 0.001, d = 1.21; IHT: MD = 3.1 (95% CI [4.8–1.5]), p < 0.001, d = 1.27; CON: MD = 2.7 (95% CI [4.4–1.1]), p < 0.001, d = 1.11) compared to T4 during all three conditions. Effort perception was higher at T24 compared to T4 only in NOR (MD = 2.1 (95% CI [3.8−0.5]), p < 0.001, d = 0.86). During IHT (MD = 1.9 (95% CI [3.5–0.2]), p = 0.009, d = 0.76) and NOR (MD = 2.9 (95% CI [4.6–1.3]), p < 0.001, d = 1.19), effort perception was higher at T40 compared to T4.

Physical strain: There was a time × condition interaction and main effect of time but no main effect of condition for physical strain. Post-hoc tests revealed no differences between conditions. Physical strain was higher at T20 (IHHT: MD = 2.5 (95% CI [4.1 to 1.0]), p < 0.001, d = 1.14; IHT: MD = 2.3 (95% CI [3.9–0.8], p < 0.001, d = 1.05; NOR: MD = 2.7 (95% CI [4.3–1.2]), p < 0.001, d = 1.23), T36 (IHHT: MD = 3.2 (95% CI [4.8–1.6), p < 0.001, d = 1.44; IHT: MD = 3.6 (95% CI [5.2–2.0]), p < 0.001, d = 1.62; CON: MD = 3.7 (95% CI [5.2–2.1]), p < 0.001, d = 1.65), and T40 (IHHT: MD = 2.0 (95% CI [3.6–0.4]), p < 0.001, d = 0.090; IHT: MD = 2.3 (95% CI [3.8–0.7]), p < 0.001, d = 1.02; CON: MD = 4.0 (95% CI [5.6–2.4]), p < 0.001, d = 1.81) compared to T4 during all three conditions. Physical strain was higher at T24 compared to T4 only in NOR (MD = 3.0 (95% CI [4.6–1.4]), p < 0.001, d = 1.35).

Affective valence: There was a time × condition interaction and a main effect of time but no main effect of condition for affective valence. However, post-hoc tests revealed no differences between or within conditions.

Arousal: There was no time x condition interaction or main effect of condition but a main effect of time for arousal. Post-hoc tests revealed that arousal was higher at T20 (MD = 1.3 (95% CI [0.5–2.0]), p < 0.001, d = 1.17), T24 (MD = 1.1 (95% CI [0.4–1.8]), p < 0.001, d = 1.03), T36 (MD = 1.6 (95% CI [0.8–2.3]), p < 0.001, d = 1.46), and T40 (MD = 1.4 (95% CI [0.8–2.3]), p < 0.001, d = 1.46) compared to T0 across conditions.

There were no interactions of main effects for motivation, conflict to continue exercise, and exercise enjoyment (see Table 1).

Muscle oxygenation

Descriptive values and results of the repeated measures ANOVAs for %ΔSmO2 and %ΔtHb are presented in Fig. 3 and Table 3, respectively.

Figure 3 Peripheral oxygen saturation (SPO2) (A), percentage changes in oxygenated hemoglobin (%ΔSmO2) (B), and total haemoglobin (%ΔtHb) (C) in the vastus lateralis muscle, as well as heart rate (D).

Physiological responses were measured immediately before (BSL) as well as at 4 (P4), 8 (P8), 12 (P12), 16 (P16), 20 (P20), 24 (P24), 28 (P28), 32 (P32), 36 (P36), and 40 min (P40) during submaximal constant-load cycling under either intermittent hypoxia-hyperoxia (IHHT), intermittent hypoxia-normoxia (IHT), or sustained normoxia (NOR). Values are presented as means and standard deviations. A number sign (#) denotes a difference (p < 0.05) compared to IHHT and IHT. An asterisk (*) denotes a difference (p < 0.05) compared to IHT. A cross (†) denotes a difference (p < 0.05) compared to NOR.

Table 3 Peripheral oxygen saturation (SpO2), percentage changes in oxygenated hemoglobin (%ΔSmO2) and total tissue hemoglobin concentration (%ΔtHb), as well as heart rate at baseline (BSL) and during exercise.

Values are presented as means ± standard deviations. The %ΔSmO2 and %ΔtHb data were averaged over the last 120 s of every 4-min period (i.e., at 4 (P4), 8 (P8), 12 (P12), 16 (P16), 20 (P20), 24 (P24), 28 (P28), 32 (P32), 36 (P36), and 40 min (P40)) and normalized to baseline (60 s) recorded immediately before exercise in a seated position on the cycle ergometer. HR data were averaged over 60 s for BSL recording and over every 4-min period during exercise. ANOVA, analysis of variance; IHHT, intermittent hypoxia-hyperoxia; IHT, hypoxia-normoxia; NOR, sustained normoxia.

Condition	BSL	P4	P8	P12	P16	P20	P24	P28	P32	P36	P40	Repeated measures ANOVA	
Time × condition interaction	Main effect of time	Main effect of condition	
SpO2 [%]	
IHHT	95.9 ± 1.3	80.4# a ± 4.7	97.7b ± 0.7	81.3# a ± 3.9	97.0b ± 1.2	81.1# a ± 4.9	96.9b ± 1.2	82.2# a ± 4.4	96.9 b ± 1.2	82.1# a ± 3.5	97.1b ± 1.0	F20, 260 = 66.180, p < 0.001, ηp2 = 0.84	F1.957, 25.444 = 151.045, p < 0.001, ηp2 = 0.92	F2, 26 = 132.163, p < 0.001, ηp2 = 0.91	
IHT	96.6 ± 1.0	77.5# a ± 8.0	93.9 ± 0.9	79.6# a ± 4.0	93.2 ± 1.2	78.5# a ± 4.0	92.7# ± 1.6	81.1# a ± 4.2	93.1# ± 1.1	81.4# a ± 4.6	92.6# ± 1.5	
NOR	96.6 ± 0.9	95.2 ± 1.2	94.4 ± 1.0	94.6 ± 0.9	93.9 ± 1.4	93.9 ± 1.4	93.6 ± 1.1	94.1 ± 1.4	93.5 ± 0.9	93.9 ± 1.7	93.7 ± 1.2	
%ΔSmO2 [%]	
IHHT		−38.8 ± 23.8	−22.1# ± 29.6	−33.2 ± 36.6	−17.9# ± 34.2	−38.1 ± 27.2	−12.7# ± 38.8	−29.3 ± 31.9	−9.2# ± 36.6	−25.7 ± 34.5	−8.1# ± 36.6	F18, 252 = 2.287, p = 0.003, ηp2 = 0.14	F2.708, 37.917 = 24.868, p < 0.001, ηp2 = 0.64	F2, 28 = 2.616, p = 0.093, ηp2 = 0.16	
IHT		−39.5 ± 21.8	−26.7 ± 20.3	−40.4 ± 23.2	−27.2 ± 23.0	−39.5 ± 24.5	−23.5 ± 21.9	−38.2 ± 22.2	−23.0 ± 21.5	−35.6 ± 22.8	−20.5# ± 22.7	
NOR		−29.1 ± 24.3	−20.9 ± 29.2	−17.5 ± 25.9	−20.3 ± 34.5	−23.7 ± 32.0	−13.0 ± 37.4	−13.9 ± 31.7	−9.1# ± 37.2	−10.5# ± 34.3	−8.3# ± 37.4	
%ΔtHb [%]	
IHHT		−1.5 ± 4.7	−2.2 ± 4.6	−1.9 ± 4.9	−1.6 ± 3.7	−1.4 ± 3.6	−1.0 ± 4.8	−1.4 ± 4.2	−0.9 ± 4.9	−1.1 ± 4.3	−0.9 ± 5.3	F18, 252 = 0.444, p = 0.977, ηp2 = 0.03	F1.600, 22.407 = 0.717, p = 0.469, ηp2 = 0.05	F2, 28 = 0.385, p = 0.684, ηp2 = 0.03	
IHT		−1.1 ± 4.3	−2.7 ± 5.7	−2.1 ± 5.9	−2.8 ± 5.7	−2.2 ± 5.9	−2.6 ± 5.4	−1.7 ± 4.4	−2.4 ± 4.2	−2.0 ± 4.4	−2.6 ± 4.2	
NOR		−1.6 ± 3.2	−1.7 ± 4.0	−1.8 ± 4.5	−1.2 ± 2.4	−1.3 ± 2.3	−0.7 ± 3.7	−0.9 ± 2.9	−0.6 ± 3.8	−1.0 ± 3.4	−0.8 ± 4.2	
Heart rate [bpm]	
IHHT	86.6 ± 7.4	136.0# ± 10.8	131.2# ± 10.7	144.6# ± 11.4	136.7# ± 11.8	149.1# ± 12.6	140.6# ± 13.1	151.4# ± 13.6	143.1# ± 16.7	154.4# ± 13.2	145.0# ± 14.5	F20, 260 = 10.344, p < 0.001, ηp2 = 0.44	F1.728, 22.464 = 222.817, p < 0.001, ηp2 = 0.95	F2, 26 = 6.513, p = 0.005, ηp2 = 0.33	
IHT	88.0 ± 10.3	138.1# ± 13.9	138.8# ± 15.1	147.9# a ± 14.8	144.7# ± 15.3	152.4# a ± 15.3	148.9# ± 16.0	155.3# a ± 15.8	151.4# ± 17.1	157.8# ± 16.4	152.9# ± 17.3	
NOR	84.2 ± 5.6	127.6# ± 10.4	133.5# ± 11.3	136.8# ± 12.9	138.7# ± 11.7	141.5# ± 12.1	142.1# ± 11.6	144.3# ± 12.3	145.3# ± 12.6	147.1# ± 13.0	146.9# ± 13.3	
Note:

# denotes a difference (p < 0.05) compared to BSL (i.e., SpO2 and HR) or P4 (i.e., %ΔSmO2, %ΔtHb).

a and b denotes a difference (p < 0.05) compared to NOR and IHT, respectively.

%ΔSmO2: There was a time × condition interaction and a main effect of time but no main effect of condition for %ΔSmO2. Post-hoc tests revealed no differences between conditions. %ΔSmO2 was lower at P8 (MD = −16.7% (95% CI [−33.2 to 0.2%]), p = 0.042, d = 0.55), P16 (MD = −20.8% (95% CI [−37.4 to −4.3%]), p < 0.001, d = 0.69), P24 (MD = −26.1% (95% CI [−42.6 to −9.6%]), p < 0.001, d = 0.86), P32 (MD = −29.6% (95% CI [−46.1.0 to −13.1%]), p < 0.001, d = 0.98), and P40 (MD = −30.6% (95% CI [47.2 to −14.1%]), p < 0.001, d = 1.01) compared to P4 during IHHT. In IHT, %ΔSmO2 was lower at P40 (MD = −19.0% (95% CI [−35.5 to −2.5%]), p = 0.004, d = 0.63) compared to P4. In NOR, %ΔSmO2 was lower at P32 (MD = −20.1% (95% CI [3.0–0.8%]), p < 0.001, d = 1.79), P36 (MD = −18.7% (95% CI [−35.2 to −2.1%]), p = 0.006, d = 0.62), and P40 (MD = −20.9% (95% CI [−37.4 to −4.4%]), p < 0.001, d = 0.69) compared to P4.

%ΔtHb: There was no time × condition interaction or main effect of time or condition for %ΔtHb.

Blood lactate concentration, peripheral oxygen saturation, and heart rate

Descriptive values and results of the repeated measures ANOVAs for BLC, SpO2, and heart rate are presented in Fig. 3 and Tables 3 and 4, respectively.

Table 4 Peripheral blood lactate concentration [mmol/l] prior to (BSL) as well as immediately (Post 0) and 10 min (Post 10) after exercise.

Data are presented as means ± standard deviations. ANOVA, analysis of variances; IHHT, intermittent hypoxia-hyperoxia; IHT, intermittent hypoxia-normoxia; NOR, sustained normoxia.

Condition	BSL	Post 0	Post 10	Repeated measures ANOVA	
Time × condition interaction	Main effect of time	Main effect of condition	
IHHT	0.38 ± 0.14	1.01# ± 0.53	0.62 ± 0.30	F = 4.934, p = 0.017, ηp2 = 0.26	F = 13.959, p = 0.001, ηp2 = 0.50	F = 3.750, p = 0.033, ηp2 = 0.28	
IHT	0.40 ± 0.17	1.43# a ± 1.17	0.95# ± 1.04	
NOR	0.40 ± 0.25	0.74 ± 0.37	0.43 ± 0.21	
Notes:

# denotes a difference (p < 0.05) compared to BSL.

a denotes a difference (p < 0.05) compared to NOR.

BLC: There was a time × condition interaction and a main effect of time and condition for BLC. Post-hoc tests revealed that BLC was higher at post 0 during IHT compared to NOR (MD = 0.7 mmol/l (95% CI [1.2–0.2 mmol/l]), p = 0.002, d = 1.16). BLC increased at post 0 during IHHT (MD = 0.6 mmol/l (95% CI [1.2–0.1 mmol/l]), p = 0.006, d = 1.07) and IHT (MD = 1.0 mmol/l (95% CI [1.6–0.5 mmol/l]), p < 0.001, d = 1.75) compared to BSL. Compared to BSL, BLC remained increased at post 10 during IHT only (MD = 0.6 mmol/l (95% CI [1.1–0.2 mmol/l]), p = 0.033, d = 0.94).

SpO2: There was a time × condition interaction and main effect of time and condition for SpO2. Post-hoc tests revealed that SpO2 was lower during IHHT and IHT compared to NOR at P4 (IHHT: MD = −14.8% (95% CI [−18.3 to −11.3%]), p < 0.001, d = 5.27; IHT: MD = −17.7% (95% CI [−21.2 to −14.2%]), p < 0.001, d = 6.31), P12 (IHHT: MD = −13.3% (95% CI [−16.8 to −9.8%]), p < 0.001, d = 4.74; IHT: MD = −14.9% (95% CI [−18.4 to −11.4%]), p < 0.001, d = 5.32), P20 (IHHT: MD = −12.7% (95% CI [−16.1 to −9.2%]), p < 0.001, d = 4.53; IHT: MD = −15.4% (95% CI [−18.8 to −11.9%]), p < 0.001, d = 5.48), P28 (IHHT: MD = −11.9% (95% CI [−15.3 to −8.4%]), p < 0.001, d = 4.23; IHT: MD = −13.0% (95% CI [−16.5 to −9.5%]), p < 0.001, d = 6.63), and P36 (IHHT: MD = −11.7% (95% CI [−15.2 to −8.2%]), p < 0.001, d = 4.18; IHT: MD = −12.5% (95% CI [−16.0 to −9.1%]), p < 0.001, d = 4.46). Compared to IHT, SpO2 was higher during IHHT at P8 (MD = 3.7% (95% CI [7.2–0.2%]), p = 0.017, d = 1.32), P16 (MD = 3.8% (95% CI [7.3–0.3%]), p = 0.017, d = 1.35), P24 (MD = 4.1% (95% CI [7.6–0.7%]), p = 0.002, d = 1.48), P32 (MD = 3.9% (95% CI [7.3–0.4%]), p = 0.009, d = 1.38), and P40 (MD = 4.6% (95% CI [8.1–1.1%]), p < 0.001, d = 1.63). Compared to BSL, SpO2 decreased at P4, P12, P20, P28, and P36 in IHHT (P4: MD = −15.5% (95% CI [−18.9 to −11.9%]), p < 0.001, d = 5.50; P12: MD = −14.6% (95% CI [−18.1 to −11.1%]), p < 0.001, d = 5.20; P20: MD = −14.7% (95% CI [−18.2 to −11.2%]), p < 0.001, d = 5.25; P28: MD = −13.6% (95% CI [−17.1 to −10.2%]), p < 0.001, d = 4.86; P36: MD = −13.7% (95% CI [−17.2 to −10.2%]), p < 0.001, d = 4.89) and IHT (P4: MD = −19.1% (95% CI [−22.6 to −15.6%]), p < 0.001, d = 6.80; P12: MD = −16.9% (95% CI [−20.4 to −13.4%]), p < 0.001, d = 6.04; P20: MD = −18.1% (95% CI = −21.6 to −14.6%]), p < 0.001, d = 6.44; P28: MD = −15.5% (95% CI [−19.0 to −12.0%]), p < 0.001, d = 5.53; P36: MD = −15.2% (95% CI [−18.7 to −11.7%]), p < 0.001, d = 5.42). During IHT, SpO2 was lower at P24 (MD = −3.9% (95% CI [−7.3 to −0.4%]), p = 0.009, d = 1.38), P32 (MD = −3.5% (95% CI [−7.0 to −0.0%]), p = 0.046, d = 1.25), and P40 (MD = −4.0% (95% CI [−7.5 to −0.5%]), p = 0.004, d = 1.43) compared to BSL. The SF-index for each hypoxic period in the IHHT and IHT condition (P4, P12, P20, P28, and P36) is shown in Supplemental Material (see Table S1).

Heart rate: There was a time × condition interaction and main effect of time and condition for heart rate. Post-hoc tests revealed that heart rate was higher during IHT compared to NOR at P12 (MD = 11.1 bpm (95% CI [21.9–0.2 bpm]), p = 0.040, d = 0.84), P20 (MD = 10.9 bpm (95% CI [21.8–0.0 bpm]), p = 0.047, d = 0.83), and P32 (MD = 11.3 bpm (95% CI [22.2–0.4 bpm]), p = 0.030, d = 0.86). Heart rate was increased during cycling compared to BSL in all conditions (see Table S2).

Discussion

The results of the present study show, for the first time, that the acute physiological reactions of young healthy males during submaximal constant-load cycling are influenced by the pattern of oxygen availability manipulation, while the perceptual responses did not differ between IHHT, IHT, and NOR. The main findings were that (i) no differences were found between conditions regarding perceptual responses, (ii) the extent of reoxygenation (i.e., higher SpO2) was greater during IHHT compared to IHT, and (iii) heart rate response and metabolic stress (i.e., increased heart rate and BLC) was higher during IHT compared to NOR.

Effects on perceptual responses

There were no differences between conditions in ratings of perceived motor fatigue, effort perception, perceived physical strain, affective valence, arousal, motivation to exercise, conflict to continue exercise, or exercise enjoyment. Therefore, the hypothesis that submaximal constant-load cycling under intermittent hypoxia would be perceived as more pleasant and enjoyable when normoxic periods are replaced by hyperoxic periods cannot be confirmed. This is in contrast to the results of Brinkmann et al. (2017) who reported a trend for a lower ratings of perceived exertion during submaximal constant-load cycling (40 min at 2.5 mmol/l BLC threshold) under intermittent hypoxia-hyperoxia (8 × 5 min, FiO2 = 0.140 and 0.300) compared to prolonged hypoxia (FiO2 = 0.140) in overweight/obese non-insulin-dependent type 2 diabetic males. The contradictory results might be due to the pattern of hypoxia, participants’ characteristics, and/or internal load of exercise. Brinkmann et al. (2017) compared two different patterns of hypoxic exposure (i.e., intermittent hypoxia-hyperoxia vs. prolonged hypoxia). Indeed, previous studies have shown differences in hematological and cardiac responses between intermittent and prolonged hypoxic exposure (Tobin, Costalat & Renshaw, 2020; Chacaroun et al., 2016). Furthermore, a previous study has shown that cycle durations during intermittent hypoxic exposures can influence participants’ perceptual responses, tending to better tolerate shorter cycle durations and a higher intra-session frequency (Hobbins et al., 2019b). Hence, it might be possible that the pattern of hypoxia can also influence perceptual responses to acute exercise. In the study by Brinkmann et al. (2017), BLC was higher after cycling under prolonged hypoxia compared to intermittent hypoxia-hyperoxia. Therefore, it could be argued that participants’ rating of perceived exertion was exacerbated in the prolonged hypoxic condition due to a greater metabolic stress (i.e., greater accumulation of metabolites). However, in the present study, the increased metabolic stress during IHT, indicated by higher BLC compared to NOR, did not seem to affect participants’ perceptual responses. This contradiction may be due to the fact that the present study examined young healthy males, whereas Brinkmann et al. (2017) included overweight/obese non-insulin-dependent type 2 diabetic participants. In this regard, obese adults typically demonstrate a reduced critical threshold (i.e., negative changes in affective valence occur as exercise intensity approaches this threshold) (Ekkekakis, Lind & Vazou, 2010) and are considered to be more perceptually sensitive to exercise-induced physiological stress compared to healthy individuals (Ekkekakis, Parfitt & Petruzzello, 2011; Hobbins et al., 2021; Ekkekakis et al., 2016). In addition, there is a difference in BLC immediately after cycling in the previous (3.84–4.65 mmol/l (Brinkmann et al., 2017)) and the current study (0.74–1.43 mmol/l), indicating that the acute interventions resulted in different internal exercise loads. Internal load has a major impact on participants’ perceived response to physical activity (Ekkekakis, Parfitt & Petruzzello, 2011). Therefore, the differences in internal exercise load could also have led to the contrast between the results of this study and the study by Brinkmann et al. (2017).

In addition, Brinkmann et al. (2017) only enquired participants’ rating of perceived exertion on a 15-point (6–20) Borg scale. However, with regard to the three-dimensional dynamical system framework of perceived motor fatigue introduced by Venhorst, Micklewright & Noakes (2018) there is a need to differentiate between the (i) perceptual-discriminatory (e.g., effort perception and perceived physical strain), (ii) affective-motivational (e.g., affective valence, arousal, and motivation), and (iii) cognitive-evaluative dimension (e.g., conflict to continue exercise) as well as its underlying constructs that underpin the individuals’ psycho-physiological state during motor tasks. For instance, effort perception can be described as the subjectively perceived effort invested into or required to continue with a task, whereas perceived physical strain is related to exercise-induced strain perception within different body parts and functions (e.g., leg discomfort, difficulty breathing, and overall discomfort) (Venhorst, Micklewright & Noakes, 2018). A previous study investigating the psycho-physiological response during self-selected continuous cycling (5 min) at a given sense of effort (three out of 10) under hypoxia (FiO2 = 0.130) and normoxia in team sport athletes found no differences in power output and quadriceps muscle activity, while the physiological response (i.e., increased heart rate and decreased SpO2) and perceived physical strain (i.e., difficulty breathing) were higher under the hypoxic compared to normoxic condition (Christian et al., 2014). This result suggests that effort perception during a given motor task is independent from the sensations that arise from peripheral afferent inputs (i.e., perceived physical strain) as described in the corollary discharge model (Pageaux, 2016). The results further imply that prolonged hypoxia negatively affects perceived physical strain caused by the motor task. This was also found during perceptually-regulated high-intensity (16 on the 6–20 Borg scale) running intervals (4 × 4 min, FiO2 = 0.150) in trained runners (Hobbins et al., 2019a). However, running velocity was decreased under hypoxia, suggesting that the reduction in FiO2 had an effect on participants’ effort perception during high-intensity interval training (Hobbins et al., 2019a). In this context, it has been shown that, among other mechanisms, environmental hypoxia can increase glucose uptake perhaps to optimize the pathways of adenosine triphosphate synthesis in face of a limited oxidative metabolism (Horscroft & Murray, 2014; Kasai et al., 2021). Indeed, anaerobic glycolysis contributes to the accumulation of metabolites, leading to a reduction in contractile muscle function (Allen, Lamb & Westerblad, 2008; Sundberg & Fitts, 2019). Consequently, greater muscle activation is needed to maintain an appropriate force output required to continue the submaximal motor task, which is associated with an increased perception of effort (Pageaux, 2016). In addition, exercise-related disturbances of the intramuscular metabolic milieu can lead to the activation of group III/IV muscle afferents, thereby increasing the perception of physical strain (e.g., muscle pain, peripheral discomfort) (Pollak et al., 2014; Mauger, 2013). Furthermore, group III/IV-mediated feedback during fatiguing motor tasks directly and indirectly impairs the output from spinal motoneurons, which in turn can affect muscle activation and thus effort perception (Taylor et al., 2016; Kennedy et al., 2015). These perceptual responses (i.e., effort perception and perceived physical strain) are thought to influence the individuals’ affective-motivational state (i.e. affective valence, arousal, and motivation) and cognitive-evaluative processes (i.e., conflict to continue exercise) (Venhorst, Micklewright & Noakes, 2018; Behrens et al., 2023). With regard to the results of the present study, it can be assumed that the hypoxia-induced metabolic perturbations elicited during 40 min of submaximal constant-load cycling in young healthy males were not sufficient to decrease contractile muscle function and/or increase group III/IV afferents’ activity to such an extent that individuals’ perceptual responses were affected. However, data indicate that this might be primarily due to the intermittent pattern of hypoxia rather than the replacement of normoxic by hyperoxic periods. Future studies should investigate whether this also applies to other exercise modalities (e.g., high-intensity interval training), hypoxic doses (e.g., FiO2, duration of periods, and/or cycle frequency), or participants (e.g., sedentary and/or overweight/obese participants or patients with metabolic disease).

Effects on physiological responses

Since oxygen saturation of arterial blood (SaO2) strongly depends on the product of FiO2 and barometric pressure in the respiratory air (i.e., partial pressure of oxygen (PO2)), hyperoxia and hypoxia are associated with an increase and a decrease in SpO2, respectively (Collins et al., 2015). Therefore, it was expected that SpO2 decreases during the hypoxic periods in the IHHT as well as IHT condition with lower values compared to NOR. Furthermore, although there were no differences during the hypoxic periods between IHHT and IHT, SpO2 was higher during the hyperoxic periods in the IHHT condition compared with the normoxic periods in the IHT condition. This indicates that the internal hypoxic intensity (i.e., SpO2) did not differ between the hypoxic cycles in both the IHHT and IHT condition, but reoxygenation was more pronounced during IHHT compared to IHT.

There is a close relationship between changes in SpO2 and V̇O2peak with the decrease in SpO2 explaining more than 70% of the reduction in V̇O2peak under acute hypoxia (up to about 4,500 m or FiO2 ≈ 0.120) (Furian, Tannheimer & Burtscher, 2022). Accordingly, it has been suggested that V̇O2peak decreases by 6.8 ± 1.4% per 1,000 m of altitude gain under normobaric hypoxia during cycling exercise (Treml et al., 2020). To the contrary, exercising under hyperoxia leads to an increase in V̇O2 and V̇O2peak at submaximal and maximal load by 4.8−10.0% and 4.0−15.0%, respectively (Sperlich et al., 2017). Consequently, studies have shown that hypoxia (FiO2 = 0.140) decreases incremental exercise test performance (i.e., maximum work during cycling) in healthy young males by about 13% (Ofner et al., 2014) while hyperoxia (FiO2 = 0.300) increases it by 4.5% (Prieur et al., 2002). Moreover, acute hypoxia reduces power output at the first and second lactate and ventilatory threshold in absolute but not in relative terms (i.e., in relation to maximal load) (Ofner et al., 2014). In the current study, the external load during exercise was individually adjusted to 60% of the participants VO2peak measured under normoxic conditions. Therefore, it is probable that the relative external load (i.e., external load in relation to the maximum load) was higher and lower during the hypoxic and hyperoxic conditions, respectively, compared to normoxia. However, for the aim of this study the external load was not adjusted to the conditions.

In contrast to SpO2, the present results indicate that a reduction or increase in FiO2 to 0.14 or 0.30, respectively, over intermittent periods of 4 min had no significant effect on %ΔSmO2. While SpO2 reflects the oxygen content expressed as percentage of the maximum oxygen capacity of the blood, SmO2 provides information on the localized equilibrium between oxygen delivery and oxygen demand (e.g., by microcirculation and mitochondrial consumption, respectively), mainly at the superficial level of the muscle (Perrey & Ferrari, 2018). SmO2 measured and calculated by mNIRS considers the relative changes in oxy- and deoxyhemoglobin as well as tHb (Feldmann, Schmitz & Erlacher, 2019). A previous study compared the acute muscle tissue oxygenation response to 30 min constant-load submaximal cycling (75% of maximal heart rate) in normoxia and continuous normobaric hypoxia (FiO2 = 13.5%, SpO2 = 75 ± 2%) (Chacaroun et al., 2019). Consistent with the results of the current study, quadriceps tissue oxygenation index did not differ between the conditions. Interestingly, oxyhemoglobin concentration was reduced with similar tHb in hypoxia compared to normoxia. This reflects a reduced oxygen delivery due to reduced SpO2 in hypoxia and may suggest that local muscle blood flow was equal between hypoxia and normoxia. A larger muscle deoxyhemoglobin concentration indicates a greater oxygen extraction (i.e., the ratio of oxygen consumption to oxygen delivery) during cycling in hypoxia. However, it must be acknowledged that the external load was adjusted to the individuals’ heart rate. Consequently, mean power output during cycling was greater in normoxia (125 ± 61 W) than in hypoxia (99 ± 49 W). Hence, the authors concluded that, on the one hand, the reduced oxygen delivery (i.e., reduced oxyhemoglobin concentration) and greater oxygen extraction (i.e., increased deoxyhemoglobin concentration) in hypoxia, on the other hand, the greater oxygen consumption in normoxia due to greater power output have led to a similar tissue oxygenation index (Chacaroun et al., 2019). Similar to this results, 30 min of submaximal constant-load cycling under hypobaric hypoxia (526 mmHg, corresponding to 3,000 m) at a relative intensity of 70% of individuals’ maximum heart rate decreased oxy- and increased deoxyhemoglobin concentration, while tHb remained unchanged compared to normoxia (Park et al., 2022). However, vastus lateralis muscle tissue oxygenation index was reduced in the hypoxic condition. With regard to the present results, although SmO2 tended to be lower during the hypoxic periods in the IHHT and IHT compared to the NOR condition, these differences were not statistically significant. Therefore, it could be assumed that the reduced SpO2 during the hypoxic periods was partially compensated by (i) central and/or peripheral mechanisms (e.g., increased heart rate, vasodilatation, and hyperemia) to maintain oxygen delivery to the working muscles (Dinenno, 2016), (ii) a duration of 4 min was too short to elicit changes in SmO2, and/or (iii) the variability of the mNIRS data was too high to observe a statistically significant differences. The extent of changes in blood oxygenation are associated, among others, with various cardiovascular, metabolic, and molecular perturbations (Burtscher et al., 2022; Sperlich et al., 2017). With respect to the cardiovascular system, skeletal muscle blood flow is increased during acute hypoxia as a consequence of local vasodilatation (Dinenno, 2016), whereas acute hyperoxia diminishes blood flow within skeletal muscle due to vasoconstriction (Welch et al., 1977). However, in the present study, %ΔtHb did not differ during both IHHT and IHT compared to NOR, indicating that blood volume in the right vastus lateralis muscle was not substantially affected by the intermittent changes in FiO2 during submaximal constant-load cycling. However, this result should be interpreted with caution as mNIRS signals per se do not measure blood flow directly, the detection area is limited to the application site, and the signal can be influenced by cutaneous blood flow (Barstow, 2019). In parallel with these vascular responses, heart rate and cardiac output typically increase under acute hypoxia (Moon et al., 2016) and decrease under acute hyperoxia (Boussuges et al., 2020). The hypoxia-induced increase in heart rate is triggered by chemoreceptor-mediated activation of the sympathetic nervous system in response to the decline in SaO2 (Burtscher et al., 2022; Hanada, Sander & González-Alonso, 2003). This is consistent with the results of the present study indicating an increased heart rate in the hypoxic periods during IHT compared with NOR. Interestingly, differences in heart rate between IHT and NOR disappeared in the last hypoxic period (i.e., after 36 min (P36)). Although speculative, this observation could be explained by i) the phenomenon of cardiovascular drift, which is characterized by a progressive increase in heart rate during prolonged submaximal constant-load exercise (Souissi et al., 2021) and ii) the decrease of individuals’ maximal heart rate under hypoxia (Mourot, 2018). Moreover, there were no differences in heart rate between IHHT and NOR in neither the hypoxic nor hyperoxic periods indicating that the replacement of normoxic by hyperoxic periods have led to a diminished cardiovascular response during the hypoxic periods. Various mechanisms have been suggested to be responsible for oxygenation-dependent cardiovascular alterations (Burtscher et al., 2022). Indeed, it has been shown that hypoxia induces an increase in adenosine plasma concentration (Saito et al., 1999), whereas hyperoxia leads to the opposite effect (Boussuges et al., 2020). Adenosine is a ubiquitous nucleoside that impacts the cardiovascular system throughout the activation of G-protein coupled receptors (i.e., A1, A2A, A2B, A3) (Burnstock, 2017). For instance, an increase in adenosine levels leads to vasodilatation (Leuenberger, Gray & Herr, 1999), whereas a decrease induces vasoconstriction and a rise in arterial blood pressure, which activates the arterial baroreflex (Demchenko et al., 2013). Afferent discharge from baroreceptors suppress sympathetic activity and reinforce parasympathetic activity, resulting in a heart rate decrease (Demchenko et al., 2013). Therefore, faster reoxygenation during hyperoxic periods might be responsible for the diminished heart rate response during IHHT, possibly due to vasomotor-evoked central signaling.

Furthermore, it has been previously suggested that PO2 affects acute skeletal muscle metabolism during exercise. When SaO2 decreases during submaximal exercise, a shift from aerobic to anaerobic metabolism has been proposed to occur (Horscroft & Murray, 2014). Therefore, glycogenolysis, glycolysis, and pyruvate production are increased under hypoxia, which would result in an excessive lactate formation leading to increases in muscle and blood lactate concentration (Kayser, 1996). This is in concordance with the results of the present study showing that BLC was increased immediately after IHHT and IHT but not after NOR. Consequently, it seems that the intermittent hypoxic periods led to an increased lactate accumulation, probably due to a shift from aerobic to anaerobic muscle metabolism. Indeed, compared with NOR, BLC was only higher after IHT but not after IHHT, which suggest that the hyperoxic periods may have affected muscle metabolism during submaximal constant-load cycling. In this regard, studies reported attenuated lactate levels in muscle and blood after 15 to 40 min of submaximal cycling (70% V̇O2peak) under hyperoxia (FiO2 = 0.600), which were associated with decreased muscle glycogenolysis, pyruvate production, and phosphocreatine utilization (Stellingwerff et al., 2005, 2006), presumably due to an increase in oxidative phosphorylation as more oxygen was available (Linossier et al., 2000). However, the mechanism underlying the perturbations in muscle metabolism are not fully understood. One possible explanation involves the sympathoadrenal system and its regulatory role in epinephrine (i.e., adrenaline) and norepinephrine (i.e., noradrenaline) production. Compared to normoxia, exercising under hypoxia (Roberts et al., 1996) enhances epinephrine and norepinephrine levels, while hyperoxia (Stellingwerff et al., 2005; Hesse et al., 1981) leads to a decrease in these catecholamines. Epinephrine binds to β-receptors in the muscle, leading to an augmented intramuscular glycogen utilization and lactate formation (Febbraio et al., 1998). Indeed, studies investigating the effect of β-blockade on glycolysis and BLC during exercise under acute hypoxia indicate that changes in lactate concentration could not be fully explained by an alteration of sympathoadrenergic control of glycolysis alone (Roberts et al., 1996; Kayser, 1996). Another explanation involves the direct effect of PO2 on substrate concentration (e.g., adenosine triphosphate, inorganic phosphate) and enzyme activity (e.g., phosphorylase, pyruvate dehydrogenase) (Parolin et al., 2000). However, discussion of the exact underlying mechanism would exceed the scope of this article. With regard to the results of the present study, it can be assumed that lactate accumulation was reduced during continuous cycling under intermittent hypoxia when normoxic reoxygenation periods were replaced by hyperoxic periods. This could be partially explained by a diminished glycogenolysis (Stellingwerff et al., 2005), a more rapid oxidation of pyruvate (Linossier et al., 2000), and ultimately, a decreased lactate production (Stellingwerff et al., 2006) and/or an increased lactate clearance (Adams, Cashman & Young, 1986) when FiO2 was elevated. Thus, in addition to a reduction in heart rate, metabolic stress is also reduced during submaximal constant-load cycling under intermittent hypoxia-hyperoxia compared to intermittent hypoxia-normoxia.

Although acute hypoxia and hyperoxia are associated with divergent alterations in oxygen availability and induce different physiological responses, both represent stimuli that elicit cellular/molecular reactions (Burtscher et al., 2022). Mitochondrial ROS production plays a pivotal role in evoking acute reactions and chronic adaptations to compensate for PO2 alterations. Both hypoxia and hyperoxia provoke ROS production (Clanton, 2007), which triggers intracellular redox signaling cascades, activating transcription factors such as hypoxia-inducible factor 1 (HIF-1) and nuclear factor erythroid-2 related factor 2 (Nrf2) (Malec et al., 2010). Activation of HIF-1 and Nrf2 induces the expression of diverse proteins mediating antioxidative and anti-inflammatory processes, cell survival, erythropoiesis, angiogenesis and tissue perfusion, iron homeostasis, as well as metabolic and cardiovascular remodeling (Semenza, 2009; He, Ru & Wen, 2020; Burtscher et al., 2023). These processes collectively adapt cells, tissues, and organs to altered oxygen concentrations that increase the organism’s tolerance to potentially harmful stimuli (i.e., hormesis phenomenon) (Sena & Chandel, 2012; Li, Yang & Sun, 2019). However, in addition to these hypoxia- or hyperoxia-induced mechanisms, several additional processes are provoked during or as a consequence of reoxygenation (Michiels, Tellier & Feron, 2016). For example, phosphoinostide-3 kinase/Akt pathway is activated during reoxygenation, which is a major trigger for HIF-1 stabilization (Martinive et al., 2009). In addition, during reoxygenation, calcium-dependent activation of NADPH-oxidase promotes ROS formation (Chen et al., 2018). Thus, an intermittent hypoxic-hyperoxic stimulus may contribute to a more robust induction of adaptive responses compared to either one condition alone or in alternation with normoxia. This is possibly due to (i) a combinatory effect of distinct reactions evoked by hypoxia vs. hyperoxia, (ii) molecular overlaps (i.e., gene transcription programs), and (iii) a reoxygenation-dependent increased accumulation of gene transcription factors (i.e., HIF-1 and ROS). These assumptions theoretically justify the combinatorial approach, which assumes that replacing normoxia with moderate hyperoxia can increase the adaptive response to an intermittent hypoxic stimulus (Burtscher et al., 2022; Mallet et al., 2020; Sazontova et al., 2012). Indeed, the combination of intermittent hypoxic and hyperoxic exposure at rest has recently been successfully applied in patients with various diseases/conditions (Behrendt et al., 2022). However, application of this approach during exercise and the resulting acute responses as well as chronic adaptions should be investigated by future studies.

Limitations

The present study has several limitations which must be considered when interpreting the results. First, a single hypoxic and hyperoxic dose (i.e., intensity (FiO2), duration, and frequency) was used for IHHT and IHT based on current recommendations (Behrendt et al., 2022; Navarrete-Opazo & Mitchell, 2014). Therefore, it is unclear whether, for example, more severe (FiO2 ≤ 0.140) or prolonged periods of hypoxia would have affected the perceptual and physiological responses. Second, a fixed FiO2 of 0.140 was administered to each participant during the hypoxic periods. However, there is considerable inter-individual variability in the response to a fixed FiO2. Therefore, a clamped SpO2 (Soo et al., 2020) instead of a fixed FiO2 approach would have been more beneficial in order to individualize the dose of hypoxia. Third, the external load during exercise (i.e., physical work performed during continuous cycling) was equal across conditions based on individuals’ performance at normoxia (60% V̇O2peak). Perhaps, further studies should investigate the psycho-physiological responses to perceptually regulated exercise under intermittent hypoxia-hyperoxia and hypoxia-normoxia (Christian et al., 2014) or V̇O2peak should be determined under identical hypoxic and hyperoxic conditions to adjust the relative external load for the respective condition. Fourth, only young healthy males were tested. Therefore, no conclusions can be drawn for females or patients with acute or chronic diseases. In a recent narrative review, Raberin et al. (2024) comprehensively discussed the effects on the physiological responses (i.e., respiratory, hemodynamic, hematological, muscle metabolism, and autonomic responses) to hypoxia and its impact on exercise for males and females. The authors concluded that females exhibit greater exercise-induced hypoxemia and diaphragm fatigue, lower sympathetic vasoconstriction and higher vasodilation, as well as higher fat, lower carbohydrate, and lower amino acid oxidation compared to males (Raberin et al., 2024). These might be related to e.g., anatomical (e.g., lung size, proportion of type I muscle fibers) and/or hormonal (e.g., estrogen level) differences between males and females. However, cardiac hemodynamics do not appear to be affected by sex differences. Currently, there is limited evidence on the acute psychophysiological responses to exercise under hypoxic and/or hyperoxic conditions with regard to sex differences, which should be investigated in future studies. Fifth, muscle activity was not measured (e.g., via surface electromyography). Since muscle activity during a motor task is related to effort perception (Pageaux, 2016), this parameter should be included in future studies. Finally, only acute psycho-physiological responses to a single 40-min submaximal constant-load cycling trial combined with intermittent hypoxia-hyperoxia and hypoxia-normoxia were examined. Whether IHHT is more effective than IHT, IHHE, or training under normoxia in improving health- or performance-related outcomes in humans, requires further investigation.

Conclusion

The present study found a higher physiological response (i.e., increased heart rate) and metabolic stress (i.e., increased BLC) during submaximal constant-load cycling in intermittent hypoxia-normoxia compared to normoxia, while intermittent hypoxia-hyperoxia did not lead to an increase in heart rate and BLC. In addition, compared to IHT, IHHT seems to improve reoxygenation indicated by a higher SpO2 during the hyperoxic periods. Although exercise-related physiological responses were influenced by the pattern of oxygen availability manipulation, there were no differences in the individuals’ perceptual responses during exercise (i.e., perceived motor fatigue, effort perception, perceived physical strain, affective valence, arousal, motivation, and conflict to continue exercise) and ratings of exercise enjoyment between conditions. Therefore, data suggest that replacing normoxic by hyperoxic reoxygenation-periods during submaximal constant-load cycling under intermittent hypoxia reduced exercise-related physiological stress but has no effect on perceptual responses and perceived exercise enjoyment in young physically active healthy males.

Supplemental Information

Supplemental Information 1 Peripheral oxygenation saturation (SpO2) to the fraction of inspired oxygen (FiO2) ratio (S/F-index) during exercise under hypoxia.

Values are presented as means ± standard deviations (SD). The coefficient of variance was calculated between-subjects for each hypoxic period and reflects the variability of the SF-index. The SpO2 data were measured after each hypoxic period (i.e., after 4 [P4], 12 [P12], 20 [P20], 28 [P28], and 36 min [P36] of continuous load cycling). IHHT, intermittent hypoxia-hyperoxia; IHT, intermittent hypoxia-normoxia.

Supplemental Information 2 Results (p-values and effect size Cohens’ d (d)) of the post hoc-tests from the analysis of variance (ANOVA) with repeated measures.

Values are presented as mean differences and 95% confidence intervals and show changes in heart rate at 4 (P4), 8 (P8), 12 (P12), 16 (P16), 20 (P20), 24 (P24), 28 (P28), 32 (P32), 36 (P36), and 40 min (P40) compared to baseline during 40 min of submaximal constant-load cycling under intermittent hypoxia-hyperoxia (IHHT), hypoxia-normoxia (IHT), and sustained normoxia (NOR).

Supplemental Information 3 Raw data set of all results.

Peripheral oxygen saturation (SpO2), percentage changes in oxygenated hemoglobin (%SmO2) and tissue hemoglobin (%tHb), and heart rate (HR) were measured prior to (BSL) and/or after every 4 min period during the 40-min cycling trial (i.e., at 4 [P4], 8 [P8], 12 [P12], 16 [P16], 20 [P20], 24 [P24], 28 [P28], 32 [P32], 36 [P36], and 40 min [P40]). Peripheral blood lactate concentration (BLC) was determined prior to (BSL) as well as immediately (post 0) and 10 min (post 10) after the 40-min cycling trial. Perceived motor fatigue, effort perception, perceived physical strain, affective valence, arousal, motivation to exercise, and conflict to continue exercise were queried after 4 (T4), 8 (T8), 20 (T20), 24 (T24), 36 (T36), and 40 min (T40) of submaximal constant-load cycling in each condition. Furthermore, perceived motor fatigue, affective valence, and arousal were also queried prior to (T0) and 10 min after (post 10) the 40-min cycling trial, whereas motivation to exercise was additionally assessed only at T0. Participants were asked to complete a revised 16-item version of the physical activity enjoyment scale (PACES) shortly after the end of cycling

BMI, body mass index; IHHT, intermittent hypoxic-hyperoxic training; IHT, intermittent hypoxic-normoxic training; NOR, continuous normoxic training; POMS, profile of mood state; VO2peak, peak oxygen consumption;

The authors would like to thank all subjects for their participation in the study.

Additional Information and Declarations

Competing Interests

Author Contributions

Human Ethics

Data Availability

The authors declare that they have no competing interests.

Tom Behrendt conceived and designed the experiments, performed the experiments, analyzed the data, prepared figures and/or tables, authored or reviewed drafts of the article, and approved the final draft.

Robert Bielitzki conceived and designed the experiments, analyzed the data, prepared figures and/or tables, authored or reviewed drafts of the article, and approved the final draft.

Martin Behrens conceived and designed the experiments, analyzed the data, prepared figures and/or tables, authored or reviewed drafts of the article, and approved the final draft.

Lina-Marie Jahns performed the experiments, authored or reviewed drafts of the article, and approved the final draft.

Malte Boersma performed the experiments, authored or reviewed drafts of the article, and approved the final draft.

Lutz Schega conceived and designed the experiments, authored or reviewed drafts of the article, and approved the final draft.

The following information was supplied relating to ethical approvals (i.e., approving body and any reference numbers):

The ethics committee of the Otto-von-Guericke University Magdeburg granted ethical approval to conduct the study (IRB approval: 68/23).

The following information was supplied regarding data availability:

The raw data measurements are available in the Supplemental File.

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
