# Peer review of "Acute psycho-physiological responses to submaximal constant-load cycling under intermittent hypoxia-hyperoxia vs. hypoxia-normoxia in young males"

_PeerJ, doi:10.7717/peerj.18027_

## Round 0.1 · original submission · Major Revisions

Dear Authors:

Thank you for submitting your manuscript titled: "Acute psycho-physiological responses to submaximal constant-load cycling under intermittent hypoxia-hyperoxia vs. hypoxia-normoxia in young males" to PeerJ. Both the reviewers and I have some relevant questions that we need clarified. Some improvements must also be undertaken in the document. Please respond to the reviewers' comments.

Kind regards

Dr. Manuel Jimenez

Reviewer 1 ·

Basic reporting

In the manuscript by Behrendt et al., the authors compared the acute perceptual and physiological responses to aerobic exercise under IHHT, IHT, and NOR conditions. They found that IHT is associated with a higher physiological response and metabolic stress, while IHHT did not lead to an increase in HR and BLC compared to NOR. And there were no differences in perceptual responses and ratings of activity enjoyment between conditions. This manuscript is clear and suitable for the readership of PeerJ.

Experimental design

The experiments were designed only for the males. It is not clear how gender may play a role in the conclusions. They authors should discuss it.

Validity of the findings

No comment

Additional comments

Line 10, the number should be 2 instead of 1.

Reviewer 2 ·

Basic reporting

The article is well-written and meets professional standards in terms of structure and data presentation. The writing is clear and precise, suitable for an academic audience specializing in the field of study.

The article includes a substantial amount of literary references, providing adequate and sufficient context about the field of study. However, some references, such as Brinkmann et al. (2017), may not be the most appropriate for the population studied in this article, as they focus on individuals with type 2 diabetes. It would be more beneficial to include studies that focus on healthy and physically active populations, allowing for a more direct and relevant comparison of the obtained results.

The structure of the article is professional and well-organized, facilitating the reader's understanding. It includes figures and tables that are clear and relevant, effectively helping to illustrate the key points of the study. These visual representations are well presented and significantly contribute to the interpretation of the data. Additionally, the article mentions the availability of raw data, which is a positive aspect for the transparency and replicability of the study, allowing other researchers to verify and build upon the presented findings.

Experimental design

Critique of Study Methodology: Determination of VO2max

Introduction

We have identified several potential improvements in the VO2max measurement protocol, aiming to enhance the accuracy and reliability of the study's findings. This critique offers suggestions based on recent literature, encouraging methodological refinements to better capture the true VO2 max and facilitate more precise study outcomes.

Interday Reliability in Untrained Subjects

In untrained subjects, the interday reliability of maximal oxygen consumption (VO2max) can exhibit significant variability, with a coefficient of variation (CV) potentially reaching up to 10% when employing data averaging intervals exceeding 30 seconds. According to Martin-Rincon and Calbet (2020), shorter data averaging intervals (15-20 seconds) are preferable for accurately capturing the true VO2max and identifying the plateau phenomenon. The current study protocol utilizes a 45-second averaging interval, which may underestimate the true VO2max value.

Verification Test Absence

The protocol also lacks a verification test. A methodology proposed by Zinner et al. (2020) suggests conducting a time-to-exhaustion test to confirm VO2max. Three minutes after the incremental test, a second test should be performed at a speed 1 km/h higher (or a 1% steeper incline if a speed of 14 km/h was not achieved) than the final stage of the incremental test.

Additional Cut-off Tests for VO2max Plateau or Peak Verification

Other cut-off tests could be employed to verify the VO2max plateau or peak. According to Knaier et al. (2019), recommended cut-off measures to reduce Type I errors in VO2max determination include RERmax ≥ 1.10 (preferably 1.15), HRmax ≥ 95% of APHR, RPEmax ≥ 19, and BLmax ≥ 8 mmol/L (preferably 10 mmol/L). The current study only used RER, one of the proposed measures, but it would have been beneficial to use at least two additional confirmation measures, such as RPE or BLmax, which were subsequently used in the experimental phase.

Impact of Hypoxia and Hyperoxia on VO2max

Hypoxia: Hypoxia (reduction in inspired O2) typically reduces VO2max due to decreased oxygen availability to the muscles.
Hyperoxia: Hyperoxia (increased inspired O2) can increase VO2max by enhancing oxygen availability and oxygen transport capacity in the blood.

Extrapolation Issues

Extrapolation Problem: Determining VO2 max in normoxia and using these values to prescribe exercise in hypoxia or hyperoxia may not be accurate, as work capacity and physiological responses can significantly vary under different oxygen conditions.
Recommendation: Ideally, VO2 max should be determined under the same conditions in which exercise will be performed (hypoxia or hyperoxia) to obtain accurate and extrapolatable measures. Alternatively, corrections based on previous studies comparing VO2max differences between normoxia and other conditions could be applied.

Individualization of Work with SpO2

The current study proposes work prescription through external load using fixed inspired oxygen fraction (FiO2) to simulate altitude. This approach does not account for individual variability in response to hypoxia, potentially leading to inadequate internal load for some subjects. Implementing a SpO2 "clamp" strategy, where FiO2 is adjusted to maintain a specific SpO2, could reduce this variability and ensure each subject receives an appropriate hypoxic dose. Soo et al. (2020) propose a SpO2/FiO2-based approach, allowing for greater individualization and precision in hypoxic dose administration.

I appreciate that this has been acknowledged in the study's limitations. However, I suggest adopting this methodology where possible. Is it feasible to reorganize the groups based on this proposed variable and redo the statistical analysis?

Your consideration of these recommendations will be invaluable in refining the study and enhancing the reliability of the results.
Referencias:
1. Knaier, R., Niemeyer, M., Wagner, J., Infanger, D., Hinrichs, T., Klenk, C., Frutig, S., Cajochen, C., & Schmidt-Trucksäss, A. (2019). Which cutoffs for secondary V̇O2max criteria are robust to diurnal variations? Medicine & Science in Sports & Exercise, 51(5), 1006-1013. doi: 10.1249/MSS.0000000000001869
2. Martin-Rincon, M., & Calbet, J. A. L. (2020). Progress update and challenges on V̇O2max testing and interpretation. Frontiers in Physiology, 11, 1070. doi: 10.3389/fphys.2020.01070
3. Soo, J., Girard, O., Ihsan, M., & Fairchild, T. (2020). The Use of the SpO2 to FiO2 Ratio to Individualize the Hypoxic Dose in Sport Science, Exercise, and Health Settings. Frontiers in Physiology, 11, 570472. doi: 10.3389/fphys.2020.570472

Validity of the findings

The article mentions the three-dimensional dynamic system model by Venhorst et al. to contextualize the results related to perceived motor fatigue. However, to maximize the value of this model, it would be beneficial to integrate it more explicitly and structurally into the data analysis. The study includes measurements of all these dimensions, and it is possible to combine them more expressly and structurally into the data analysis. Applying the model in a multidimensional manner, considering and analyzing the three dimensions proposed by Venhorst et al., would allow for a richer and more robust interpretation of the study's results. To achieve this, we should integrate nonlinear dynamic analyses (such as time series analysis or principal component analysis).

On the other hand in the discussion (520-530)

The lack of differences in %SmO2 of the vastus lateralis suggests that, at the muscular level, oxygenation may be regulated by factors other than arterial SpO2 variation. This could also indicate a local adaptation to hypoxic and normoxic conditions that are not necessarily reflected in global SpO2 measures.

However, due to methodological limitations, the lack of individualization of hypoxia in the current study might have concealed significant differences in muscle oxygenation. Adopting methods that consider individual variability, such as the SpO2/FiO2 index, could enhance the precision and relevance of findings in future research.

---

## Round 0.2 · accepted · Accept

Dear Authors:

Your manuscript - Acute psycho-physiological responses to submaximal constant-load cycling under intermittent hypoxia-hyperoxia vs. hypoxia-normoxia in young males - has been Accepted for publication. Congratulations!

Regards


Dr. Manuel Jiménez

Reviewer 2 ·

Basic reporting

no comment

Experimental design

About the topic of Determination of VO2max.

Thank you for your clear feedback. Please consider incorporating these concerns into the study's limitations to ensure scientific rigor. Best regards

Validity of the findings

no comment

Additional comments

Individualization of Work with SpO2
Thanks for the text attempting to clarify the issue. It could be interesting in the following papers to analyze if it's possible to use SPO2 even under intermittent hypoxic conditions.